# Towards Text Generation with Adversarially Learned Neural Outlines

**Sandeep Subramanian**[1,2,4][*]**Sai Rajeswar**[1,2,5]**, Alessandro Sordoni**[4]**,**
**Adam Trischler**[4]**, Aaron Courville**[1,2,6]**, Christopher Pal**[1,3,5]
[1]Montréal Institute for Learning Algorithms, [2]Université de Montréal,
[3]École Polytechnique de Montréal, [4]Microsoft Research Montréal,
[5]Element AI, Montréal, [6]CIFAR Fellow
[2]{sandeep.subramanian.1,sai.rajeswar.mudumba,aaron.courville}@umontreal.ca
[3]christopher.pal@polymtl.ca
[4]{alsordon,adam.trischler}@microsoft.com

## Abstract

Recent progress in deep generative models has been fueled by two paradigms – autoregressive and adversarial models. We propose a combination of both approaches with the goal of learning generative models of text. Our method first produces a high-level sentence outline and then generates words sequentially, conditioning on both the outline and the previous outputs. We generate outlines with an adversarial model trained to approximate the distribution of sentences in a latent space induced by general-purpose sentence encoders. This provides strong, informative conditioning for the autoregressive stage. Our quantitative evaluations suggests that conditioning information from generated outlines is able to guide the autoregressive model to produce realistic samples, comparable to maximum-likelihood trained language models, even at high temperatures with multinomial sampling. Qualitative results also demonstrate that this generative procedure yields natural-looking sentences and interpolations.

## 1 Introduction

Deep neural networks are powerful tools for modeling sequential data [36, 54, 24]. Tractable maximum-likelihood (MLE) training of these models typically involves factorizing the joint distribution over random variables into a product of conditional distributions that models the one-step-ahead probability in the sequence via the chain rule. Each conditional is then modeled by an expressive family of functions, such as neural networks. These models have been successful in a variety of tasks. However, the only source of variation is modeled in the conditional output probability at every step: there is limited capacity for capturing the higher-level structure likely present in natural text and other sequential data (e.g., through a hierarchical generation process [46]).

Variational Autoencoders (VAE) [28] provide a tractable method to train hierarchical latent-variable generative models. In the context of text data, latent variables may assume the role of sentence representations that govern a lower-level generation process, thus facilitating controlled generation of text. However, VAEs for text are notoriously hard to train when combined with powerful auto-regressive decoders [5, 18, 47]. This is due to the "posterior collapse" problem: the model ends up relying solely on the auto-regressive properties of the decoder while ignoring the latent variables, which become uninformative. This phenomenon is partly a consequence of the restrictive assumptions on the parametric form of the posterior and prior approximations, usually modeled as simple diagonal Gaussian distributions.

---

[*]Work done while author was an intern at Microsoft Research Montreal

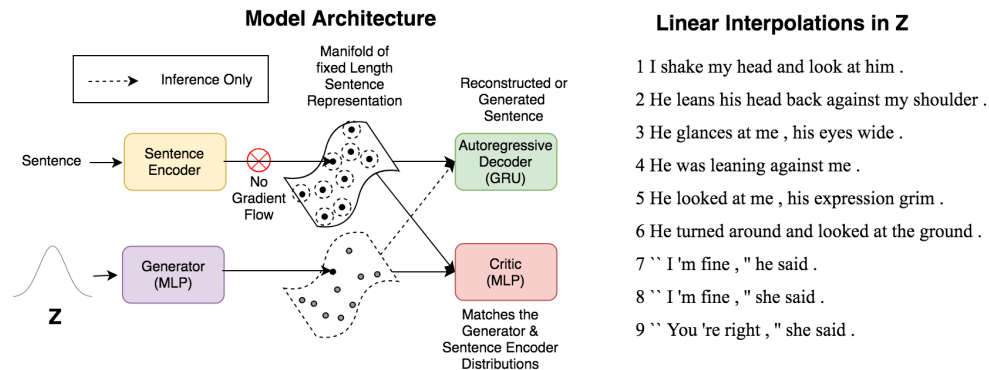

Figure 1: Overall Model Architecture. (Left) A GAN setup that is trained to model the distribution of fixed-length sentence vectors. A minimal amount of noise, indicated by a small Gaussian ball, is injected to every point on the data mainfold. (Right) Samples produced from our generative process by interpolating linearly between two points sampled at random from the input noise space of the generator.

General-purpose sentence encoders have been shown to produce representations that are useful across a wide range of natural language processing (NLP) tasks [29, 12, 51]. Such models seem to capture syntactic and semantic properties of text in self-contained vector representations [51, 10]. Although these representations have been used successfully in downstream tasks, their usefulness for text generation *per se* has not yet been thoroughly investigated, to the best of our knowledge.

In this paper, we study how pre-trained sentence representations, obtained by general purpose sentence encoders, can be leveraged to train better models for text generation. In particular, we interpret the set of sentence embeddings obtained from training on large text corpora as samples from an expressive and unknown prior distribution over a continuous space. We model this distribution using a Generative Adversarial Network (GAN), which enables us to build a fully generative model of sentences as follows: we first sample a sentence embedding from the GAN generator, then decode it to the observed space (words) using a conditional GRU-based language model (which we refer to as the decoder in the rest of this work). The sentence embeddings produced by the learned generator can be seen as "neural outlines" that provide high-level information to the decoder, which in turn can generate many possible sentences given a single latent representation. The conditioning information effectively guides the decoder to a smaller space of possible generations.

The takeaways from our work are:

- We propose the use of fixed-length representations induced by general-purpose sentence encoders for training generative models of text, and demonstrate their potential both qualitatively and quantitatively.

- We extend our model to conditional text generation. Specifically, we train a conditional GAN that learns to transform a given hypothesis representation in the sentence embedding space into a premise embedding that satisfies a specified entailment relationship.

- We propose a gradient-based technique to generate meaningful interpolations between two given sentence embeddings. This technique may navigate around "holes" in the data manifold by moving in areas of high data density, using gradient signals from the decoder. Qualitatively, the obtained interpolations appear more natural than those obtained by linear interpolation when the distribution (in our case, the induced sentence embedding distribution) doesn't have a simple parametric form such as a Gaussian.

## 2  Related Work

Our work is similar in spirit to the line of work that employs adversarial training on a latent representation of data, such as Adversarial Autoendoers (AAE) [33], Wasserstein Autoencoders (WAE) [53] and Adversarially Regularized Autoencoders (ARAE) [26]. AAEs and WAEs are similar to Variational Autoencoders (VAE) [28] in that they learn an encoder that produces an approximate latent posterior distribution $q_\phi(z|x)$, and a decoder $p_\theta(x|z)$ that is trained to minimize the reconstruction error of $x$. They differ in the way they are regularized. VAEs penalize the discrepancy between the approximate posterior and a prior distribution $D_{\mathrm{KL}}(q_\phi(z|x)\|p(z))$, which controls the tightness of the variational bound on the data log-likelihood. AAEs and WAEs, on the other hand, adversarially match the aggregated or marginal posterior, $q_\phi(z)$, with a fixed prior distribution that shapes the posterior into what is typically a simple distribution, such as a Gaussian.

ARAEs train, by means of a critic, a flexible prior distribution regularizing the sentence representations obtained by an auto-encoder. In contrast, we provide evidence that assuming a uniform prior distribution during the training of sentence representations and fitting a flexible prior *a posteriori* over the obtained representations yields better performance than learning both jointly, and helps us scale to longer sequences, larger vocabularies, and richer datasets.

Recent successes of generative adversarial networks in the image domain [25] have motivated their use in modeling discrete sequences like text. However, discreteness introduces problems in end-to-end training of the generator. Policy gradient techniques [57] are one way to circumvent this problem, but typically require maximum-likelihood pre-training [58, 6], as do actor-critic methods [17, 15]. Gumbel-softmax based approaches have also proven useful in the restricted setting of short sequence lengths and small output vocabulary sizes [31]. Approaches without maximum-likelihood pre-training that operate on continuous softmax outputs from the generator such as [19, 43, 42] have also shown promise, but apply mostly in artificial settings. Adversarial training on the continuous hidden states of an RNN was used by [48] for unaligned style transfer and unsupervised machine translation [32]. While our approach can potentially be applied to unsupervised machine translation, we believe that high quality sentence representations and resources to learn them in languages other than English, have only recently been explored [13].

In this work, we argue that generative adversarial training on latent representations of a sentence not only alleviates the non-differentiability issue of regular GAN training on discrete sequences, but also eases the learning process by instead modeling an already smoothed manifold of the underlying discrete data distribution. We also simultaneously reap the benefits of a "sequence-level" training objective while somewhat side-stepping the temporal credit assignment problem.

## 3  Approach

In this section we discuss the building blocks of our generative model. Our model consists of two distinct and independently trainable components: (1) a generative adversarial network trained to match the distribution of the fixed-length vector representations induced by general purpose sentence encoders, and (2) a conditional RNN language model trained with maximum-likelihood to reconstruct the words in a sentence given its vector representation. The overall architecture is presented in Fig. 1.

### 3.1  General Purpose Sentence Encoders

With the success of models that learn distributed representations of words in an unsupervised manner, there has been a recent focus on building models that learn fixed-length vector representations of whole sentences that are "general purpose" enough to be useful as black-box features across a wide range of downstream Natural Language Processing tasks. Extensions of the skip-gram model to sentences [29], sequential de-noising autoencoders [22], features learned from Natural Language Inference models [12], and large-scale multi-task models that are trained with multiple objectives [51] have been shown to learn useful [11, 56] fixed-length representations of text. They also encode several characteristics of a sentence faithfully, including word order, content, and length [10, 1, 51]. This is important, since we want to reliably reconstruct the contents of a sentence from its vector representation. In this work, we will use the pre-trained sentence encoder from [51], which consists of an embedding look-up table learned from scratch and a single layer GRU [9] with 2048 hidden units. We use the last hidden state of the GRU to create a compressed representation. Thus, each

sentence $x$ is represented by a vector $E(x) = h_x \in \mathbb{R}^{2048}$, where $E$ denotes our general-purpose sentence encoder.

## 3.2 Generative Adversarial Training

Generative Adversarial networks are a family of implicit generative models that formulate the learning process as a two player minimax game between a generator and discriminator/critic [16]: the critic is trained to distinguish samples of the true data distribution from those of the generator distribution. The generator is trained to "fool" the critic by approximating the data distribution, given samples from a much simpler distribution, e.g., a Gaussian. The discriminator $D$ or critic $f_w$ and the generator $G$ are typically parameterized by neural networks. In our setting, the data distribution is the distribution $P(h_x)$ of sentence embeddings, $h_x = E(x)$, obtained by applying sentence encoder $E$ to samples $x \sim P_D$ from a dataset $D$. Specifically, our critic and generator are trained with objective

$$\min_G \max_D V(D, G) = \mathbb{E}_{x \sim P_D}[\log D(E(x))] + \mathbb{E}_{z \sim P(z)}[\log(1 - D(G(z)))]$$

In presenting the Wasserstein GAN, [3] argue that while KL divergence is a good divergence for low-dimensional data, it is often too strong in the high-dimensional settings we are often interested in modeling. We found training to be more stable using the 1-Wasserstein distance between two distributions. We follow the setup of [19], which we found leads to more effective and robust training.

## 3.3 Decoding Generated Sentence Vectors to Words

Given generated or real sentences in their fixed-length latent space, we would like to train a model that maps these vectors back to their respective sentences. To this end, we train a conditional GRU language model that, for each sentence $x$, conditions on the sentence vector representation, $h_x$, as well as previously generated words at each step to reconstruct $x$. We parameterize this decoder with an embedding lookup-table, a single layer GRU, and an affine transformation matrix that maps the GRU hidden states to the space of possible output tokens. The GRU is fed the sentence vector at every time step as described in [51]. The decoder is trained with maximum-likelihood.

Given that $E$ is a deterministic function of $x \sim P_D$, where $P_D$ is a discrete distribution in the case of text data, the distribution $P(h_x)$ is a discrete distribution embedded in a continuous space. Our sentence embedding generator approximates the high-dimensional discrete distribution $P(h_x)$ with a continuous distribution $G(z)$. Therefore, samples from $G(z)$ will likely produce data points that are off-manifold; this may lead to unexpected behavior in the decoder, since it has been trained only on samples from the true distribution $P(h_x)$. To encourage better generalization for samples from $G(z)$, during the training of the decoder we smooth the distribution $P(h_x)$ by adding a small amount of additive isotropic Gaussian noise to each vector $h_x$.

In Table 1, we demonstrate the impact of noise on the reconstruction BLEU scores and sample quality. We notice that the amount of noise injected introduces a trade-off between reconstruction and sample quality. Specifically, as we increase the amount of noise the reconstruction BLEU score decreases but *samples* from a GAN start looking qualitatively better. This aligns with observations made in [14], where VAEs trained with a variance of $\sim 0.1$ had crisp reconstructions but poor sample quality, while those with a variance of $1.0$ had blurry reconstructions but better sample quality.

| Noise | BLEU-4 | Samples |
|-------|--------|---------|
| 0.00 | 64.54 | the young men are playing volleyball in the ball . |
| 0.07 | 53.12 | - |
| 0.12 | 48.60 | the young child is playing soccer . |
| 0.17 | 41.16 | - |
| 0.20 | 37.51 | a young child is playing with a ball . |

Table 1: The impact of noise on *reconstruction* quality, measured by tokenized BLEU-4 scores on SNLI [4]. We also show samples from decoders trained with different amounts of noise but always conditioned on the **same** sentence vector generated from a GAN.

The injection of noise into neural models has been explored in many contexts [2, 55, 40, 50, 41] and has been shown to have important implications in unsupervised representation learning. Theoretical and empirical arguments from Denoising Autoencoders (DAEs) [55] show that the addition of noise leads to robust features that are insensitive to small perturbations to examples on the data manifold. Contractive Autoencoders (CAEs) [44] impose an explicit invariance of the hidden representations to small perturbations in the input by penalizing the norm of the Jacobian of hidden representations with respect to the inputs. This was shown to be equivalent to

adding additive Gaussian noise to the hidden representations [41], where the variance of the noise is proportional to the contraction penalty in CAEs. Although this was proved to be true only for feedforward autoencoders, we believe this has a similar effect on sequential models like the ones we use in this work.

### 3.4 Model Architecture

In all experiments, we use the sentence encoder from [51]. In our generator and discriminator, we use 5-layer MLPs with 1024 hidden dimensions and leaky ReLU activation functions. We use the WGAN-GP formulation [19] in all experiments, with 5 discriminator updates for every generator update and a gradient penalty coefficient of 10. Our decoder architecture is identical to the multi-task decoders used in [51]. We trained all models with the Adam [27] stochastic optimization algorithm with a learning rate of 2e-4 and $\beta_1 = 0.5, \beta_2 = 0.9$. We used a noise radius of 0.12 for experiments involving SNLI and 0.2 for others.

## 4 Walking the Latent Space via Gradient-Based Optimization

We explore three different techniques to produce interpolations between sentences. The first, which is presented in Table 5, interpolates linearly in the *input noise space of the GAN generator*. The second and third techniques, which are presented in this section, interpolate between two *given* sentences in the *sentence encoder latent space* linearly or via gradient-based optimization. We show that the gradient-based method works on high-dimensional continuous representations of text that are more expressive than the Gaussian typically used in VAEs. We exploit the fact that we have continuous representations on which we can take gradient steps to iteratively transform one sentence into the other. We use our decoder that maps sentence representations into words to provide the gradient signal to move the sentence representation of the first sentence towards the second.

Specifically, given two sentences $x_1$ and $x_2$, we formulate the interpolation problem as an optimization that iteratively transforms $x_1$ into $x_2$ by taking gradient steps from $h_{x1}$ in order to reconstruct $x_2$. We start the optimization process at $h_0 = h_{x_1}$ and take gradient steps as follows

$$h_t = h_{t-1} + \alpha \nabla_{h_{t-1}} \log P(x_2|h_{t-1})$$

$\log P(x_2|h_{t-1})$ is given by our decoder described in Section 3.3. At every step of the optimization process, we can run the sentence representation $h_t$ through our decoder to produce an output sentence. Unlike linear interpolations, this procedure is not guaranteed to be symmetric, i.e., the interpolation path between $x_1$ and $x_2$ might not be the same as the path between $x_2$ and $x_1$. Sample interpolations using this technique are presented in Table 2.

| of course, i had already made coffee and she headed right for the pot. | of course, i had already made coffee and she headed right for the pot. |
|---|---|
| of course, she had already made coffee. | of course, i had already made coffee and she headed right for the pot. |
| colin had already made a pot of coffee. | of course, i had already made coffee. |
| colin pulled out the coffee pot . | i had a lot of things to do . |
| colin pulled out the file . | colin pulled the file out of his pocket . |
| colin pulled out the myers file . | colin colin pulled colin out of the colin . |
| **colin pulled out the myers file .** | **colin pulled out the myers file .** |
| **" my mother struggled to make ends meet when i was a child .** | **" my mother struggled to make ends meet when i was a child .** |
| " my mother struggled to make ends meet . | " my mother struggled to make ends meet when i was a child . |
| " my mother would make ends meet . | " my mother struggled to make ends meet . |
| " my mother would 've loved you too . | i 'm so sorry , " i said . |
| you would 've loved her mother 's child . | i love you , i love you . " |
| you would 've loved her . " | you loved him , would n't you ? " |
| **you would 've loved her . "** | **you would 've loved her . "** |

Table 2: Interpolations using gradient-based optimization (Left). Corresponding linear interpolations directly in the sentence representation space (Right). Two randomly selected sentences from the BookCorpus are in bold.

## 5 Experiments & Results

To evaluate our generative model, we set up unconditional and conditional English *sentence* generation experiments. We consider the SNLI [4], BookCorpus [59] and WMT15 (English fraction of the En-De parallel corpus) datasets in unconditional text generation experiments. For conditional generation

experiments, we consider two different tasks: (1) to generate a premise sentence that satisfies a particular relationship (entailment/contradiction/neutral) with a given hypothesis sentence from the SNLI dataset, similar to [49, 30]; and (2) a synthetic/toy task of generating captions (without the image) given a particular binary sentiment from the SentiCap dataset [34] and product reviews from Amazon [48].

## 5.1 Unconditional Sentence Generation

In all settings, we partition the dataset into equally sized halves, one on which we train our generative model (GAN & Decoder) and the other for evaluation. The large holdout set is a result of the nature of our evaluation setup, described in the next section. In Tables 4 and 5, we present samples and interpolations produced from our model on three datasets. Additionally, we also trained an InfoGAN model [8] with a 10 dimensional latent categorical variable on the BookCorpus. As shown in Appendix Table 5, the latent variable is able to pick up on some of the frequent factors of variation in the dataset such as the subject of the sentence, punctuation, quotes etc.

In Table 6, we compare unconditional samples from ARAEs[2], a state-of-the-art LSTM language model[3] [35] and our model. We experimented with different hyperparameter configurations for the ARAE to increase model capacity but found that the default parameters gave us the best results. For [35], we use the default hyper-parameters without Averaged SGD since we noticed that it didn't have an impact on results.

| Dataset | Sentences | Tokens |
|---|---|---|
| SNLI | 1.1M | 12.2M |
| BookCorpus | 12M | 159.5M |
| WMT15 | 4.5M | 117.2M |

Table 3: Dataset Statistics

Our model with beam search is competitive with the WD-LSTM at low temperatures (0.5) in terms of sample quality and diversity, but is able to maintain quality even at high temperatures (1.0). We also outperform the ARAE on the benchmarks. All experiments were performed with a vocabulary size of 80,000 words and a maximum sequence length of 50, except for the ARAE model trained on SNLI where we used the pre-trained models provided in the official code repository.

| | |
|---|---|
| 1 | the room was nicely decorated and the two of them were very comfortable and the bathroom was fantastic . |
| 2 | all of the information was gathered from the police or the court of justice in the united states . |
| 3 | we are working with the elders to tell the story of the ancient egyptian stories of the past . |
| 4 | all of our doctors , nurses , and other health care providers have been waiting for me . |
| 5 | and this is why it is so important that the health care system be fully understood . |
| 6 | this is going to be a good way to get a glimpse of the new york city council . |
| 7 | " what 's going on with you ? " |
| 8 | i shook my head , not trusting myself . |
| 9 | i was too tired to think about it . |
| 10 | " yes , " he said , nodding . |
| 1 | in the mid-1980s , he was appointed as a member of the court of human rights in afghanistan . |
| 2 | do you have any other ideas about cooperation between the european union and other countries in the world ? |
| 3 | secondly , i am not happy to see that the countries of the european union are in agreement . |
| 4 | the main objective of this study is to promote the development of a more comprehensive and accessible information society . |
| 5 | we've been looking forward to welcoming you to the beach , with a view of the sea . |
| 6 | but it is clear that the west and east of the country are not yet fully committed . |
| 7 | i would like to point out to the house that there are some amendments to the fisheries act . |
| 8 | health and education , research and development are a major factor in the development of health education programs . |
| 9 | i therefore ask the commission to cooperate fully with the commission and to parliament to approve this report . |
| 10 | i hope that the next step will be to ensure that this agreement is maintained in the eu . |

Table 4: Generated samples from our model trained on the BookCorpus (top) and WMT15 (bottom)

## 5.2 Towards Quantitatively Evaluating Unconditional Sentence Samples

Evaluating implicit generative models such as GANs is still an active area of research, with several pieces of work focusing on evaluating image samples [45, 21]. In this section, we revisit the evaluation method that was originally proposed in [16], of fitting a non-parametric kernel density estimator (KDE) on the samples produced by a GAN and then evaluating the likelihood of real examples under this KDE. As pointed out in [52], KDEs seldom do a good job of capturing the underlying density, since they do not scale well with data. However, when the underlying data distribution is discrete, count-based non-parametric models such as smoothed n-gram language models [20] are extremely

| 1 | " you 're human , " she said softly . | 1 | a girl on a stage holding a guitar . |
|---|---|---|---|
| 2 | " you 're a vampire , " she said . | 2 | a girl on a stage holding a scythe . |
| 3 | " you 're a vampire , " she said . | 3 | the girl in the sweat shirt plays a guitar on a stage . |
| 4 | " you 're a b**ch , " she said . | 4 | the girl in the sweat vest is reading a newspaper . |
| 5 | " you 're angry , " he said flatly . | 5 | a woman in a white shirt is taking a shortcut . |
| 6 | " you 're a jerk , " he said dryly . | 6 | a woman in a white shirt is holding a scythe . |
| 7 | " you 're not going to let me out ? " | 7 | a woman is playing the sax . |
| 8 | " i do n't know why you 're here ? " | 8 | a man is in the firehouse . |
| 9 | " i do n't know why you 're here ? " | 9 | a man is in the firehouse . |
| 10 | " you do n't know what to do ? " | 10 | two men are in an enclosed room . |

| 1 | it is therefore quite impossible to separate the various provisions of the single market with regard to quotas . |
|---|---|
| 2 | i do not therefore think that it is necessary to continue with a different view of the commission . |
| 3 | i am therefore very sensitive to the issue of women in different member states and the social repercussions . |
| 4 | therefore , i am very pleased with the report on women and their social rights in the member states . |
| 5 | i am also very pleased to have the opportunity to discuss with the european parliament on this matter . |
| 6 | i would also like to take the opportunity to comment on the issue of the european economic partnership . |
| 7 | i would not like to mention the commission 's statement on the issue of the council 's statement . |
| 8 | i do not want to answer any of the commissioner 's questions to the commission . |
| 9 | mr president , i am not going to reply to mr santer 's statement on the lisbon strategy . |
| 10 | mr president . |

Table 5: Linear interpolations along two randomly sampled points in the input space of the GAN generator on the BookCorpus (top left), SNLI (top right), and WMT15 (bottom). Points along the line between the two points are transformed into sentence vectors via the generator and then decoded with beam search.

| Dataset | ARAE | | | | | | WDLSTM | | | | Ours | | | | | |
|---|---|---|---|---|---|---|---|---|---|---|---|---|---|---|---|---|
| | FPPL | | | RPPL | | | FPPL | | RPPL | | FPPL | | | RPPL | | |
| | 0.5 | 1.0 | B=1 | 0.5 | 1.0 | B=1 | 0.5 | 1.0 | 0.5 | 1.0 | 0.5 | 1.0 | B=5 | 0.5 | 1.0 | B=5 |
| BookCorpus | 389.6 | 555.6 | 364.2 | 209.2 | 206.2 | 213.3 | 9.4 | 185.2 | 280.7 | 137.2 | 25.5 | 66.6 | 10.5 | 220.4 | 152.8 | 250.9 |
| WMT15 | 448.7 | 965.1 | 385.8 | 476.2 | 378.7 | 626.3 | 21.4 | 369.0 | 528.9 | 250.5 | 105.5 | 212.9 | 19.9 | 350.5 | 254.1 | 373.2 |
| SNLI | 67.5 | 109.1 | 62.0 | 54.8 | 54.0 | 59.9 | 5.9 | 57.0 | 86.8 | 34.5 | 18.6 | 35.6 | 15.3 | 90.8 | 49.5 | 59.8 |

Table 6: Quantitative evaluation of sample quality from ARAE, WD-LSTM and our model. We report the FPPL and RPPL from a KN smoothed 5-gram language modeled trained on a distinct but large subset of the data. We also report FPPLs and RPPLs on samples generated with multinomial sampling with temperatures of 0.5 and 1.0, as well as deterministic decoding with beam search (B). Note that deterministic decoding is not suitable for the WDLSTM since there needs to be some stochasticity to produce diverse samples.

| WMT (Temperature 1.0) | Grammaticality | Topicality | Overall |
|---|---|---|---|
| Ours (Temperature 1.0) | **46.19%** | **48.73%** | **63.95%** |
| WD-LSTM (Temperature 1.0) | 28.90% | 25.88% | 36.05% |
| No preference | 24.91% | 25.39% | - |
| **WMT (Beam Search vs Temperature 0.5)** | | | |
| Ours (Beam Search) | 20.00% | 15.20% | **53.20%** |
| WD-LSTM (Temperature 0.5) | 19.30% | 24.80% | 46.80% |
| No preference | **60.7%** | **60.00%** | - |
| **BookCorpus (Temperature 1.0)** | | | |
| Ours (Temperature 1.0) | **50.75%** | **54.27%** | **70.35%** |
| WD-LSTM (Temperature 1.0) | 19.59% | 23.61% | 29.65% |
| No preference | 29.66% | 22.12% | - |
| **BookCorpus (Beam Search vs Temperature 0.5)** | | | |
| Ours (Beam Search) | 22.68% | 35.29% | **57.28%** |
| WD-LSTM (Temperature 0.5) | 25.21% | 17.64% | 42.72% |
| No preference | **52.11%** | **47.07%** | - |

Table 7: Human evaluation of grammaticality, topicality and overall quality for sentences generated by our model and the WD-LSTM.

fast to train and produce reasonable density estimates which can be used as weak proxies for the real and model data densities.

Further, we identify a few simple statistics about generated samples to act as proxies for sample quality and diversity. Specifically, we report the number of unique n-grams as a proxy for the latter and the fraction of "valid" n-grams as a proxy for former (see Supplementary Table 1). We also attempt to analyze the extent to which these models overfit to examples in the training set (see Supplementary Table 2 and details in Supplementary section 1.1).

| Label | Given Hypothesis | Generated Premise |
|---|---|---|
| E | the woman is very happy . | a lady wearing a blue white shirt is laughing . |
| C | no one is dancing . | a group of people playing guitar hero on a stage . |
| N | the man is reading the sportspage . | a man in a white shirt is sitting in a recliner . |
| E | a man is in a black shirt | a man in a black shirt stands in front of a store while a man in a blue hat and white shirt stands beside him . |
| C | an old woman has no jacket . | a woman with a white hat and jacket is playing with a girl in a red jacket . |
| N | a person is waiting for a train . | person in white and black hat standing in front of a train track . |

Table 8: Samples from our GAN trained to conditionally transform a given hypothesis and a label as either (E)ntailment, (C)ontradiction or (N)eutral into a premise that satisfies the specified relationship

Let $P$ be the data distribution and $Q$ be our model's distribution. Since $P$ is unknown and $Q$ is an implicit generative model, we do not have access to either of the underlying data generating distributions, only samples from them. Nevertheless, we'd like to find an evaluation criterion that measures an (approximate) measure of similarity between these distributions.

To do so, we fit a non-parametric Kneiser-Ney smoothed 5-gram language model [20] to samples from $P$ and $Q$ and denote the resulting density models as $\hat{P}$ and $\hat{Q}$. Using these, we formulate two complementary evaluation criteria, the *Forward* and *Reverse* perplexities (FPPL, RPPL). These correspond to *approximations* of $H(Q, P)$ and $H(P, Q)$, respectively. It is straightforward to see that this is the case by substituting $Q$ with $\hat{Q}$ $H(P, Q) \simeq - \mathop{\mathbb{E}}_{x \sim P} \log \hat{Q}(x)$ and $P$ with $\hat{P}$ $H(Q, P) \simeq - \mathop{\mathbb{E}}_{x \sim Q} \log \hat{P}(x)$. We use the forward and reverse terminologies in the *opposite* way researchers refer to the forward and reverse KL divergences in an optimization setting when $P$ is the true distribution we'd like to approximate with $Q$. This is to be consistent with [26].

Computationally, the RPPL is equivalent to training a KN 5-gram LM on the samples from our model and reporting perplexities on the real data, while FPPL involves the opposite. Note that $H(P, Q)$ and $H(Q, P)$ differ in the order of arguments in the KL-divergence, with the latter being more sensitive to sample quality and the former to a balance between diversity and quality.

Finally, we also carried out human evaluations to compare samples from our model and the WD-LSTM in an A/B test. Annotators are presented with two samples - one from our model and one from the WD-LSTM (with the presentation order random each time) and asked to pick which they prefer along three dimensions: grammaticality, topicality, and overall quality. This protocol is identical to [15] who also evaluate unconditional text generation samples. Annotators are allowed to rate two samples as having equal grammaticality and topicality but not overall quality. We collected a total of 1,094 annotations from 16 annotators. In Table 5.1, we report results for comparisons across the WMT and BookCorpus datasets with different sampling parameters: Temp=1.0 and Temp=0.5 vs. beam search. We compare our beam search variant with the WD-LSTM at Temp=0.5 since we found it to be the most similar to beam search in trading off RPPL and FPPL. Every entry in the table corresponds to the percentage of annotations where annotators preferred a sample from a particular model. Our model does consistently better at high temperatures, while being comparable to the WD-LSTM at low temperatures.

## 5.3 Conditional Text Generation

In most real world settings, we are interested in conditional rather than unconditional text generation. Conditional GANs [37] have proven extremely powerful in this context for generating images conditioned on certain discrete attributes. In this work, we explore the relatively simple and artificial task of generating sentences conditioned on binary sentiment labels from the SentiCap and Amazon Review datasets. While true sentiment is certainly more nuanced, the binary setting can serve as a simple testbed for initial experiments with such techniques. Using the SNLI dataset, we also explore the task of generating a premise sentence conditioned on a given hypothesis plus a

| Method | Accuracy |
|---|---|
| Random | 41.1% |
| Baseline-Seq2Seq (Mean) | 59.6% |
| Baseline-Seq2Seq (MOSM) | 62.6% |
| Shen et al. Mean (N=1) | 62.4% |
| Shen et al. MOSM (N=1) | 75.9% |
| Ours | 70.8% |

Table 9: NLI classification accuracies of generated samples on the test set evaluated by the ESIM model [7]. All results except ours were obtained from Shen et. al [49]. Our model also had an FPPL of 15.01, which is comparable to results in Table 6

label that specifies a relationship between them [49]. We believe that transforming hypotheses into premises in SNLI is a harder problem than the inverse since it requires filling in extra details rather than removing them.

In both these sets of experiments, we use a conditional GAN with our generators and discriminators as MLPs that use conditioning information in their input layers. Conditioning information for sentiment and NLI labels is learned via a 128-dimensional embedding layer updated only by the discriminator. For premise generation, the MLPs are also presented with the sentence representation corresponding to the given hypothesis. We present quantitative evaluations of both models in Tables 9 and 10, evaluated using a combination of pre-trained classifiers and sample fluency evaluations using FPPL. On SNLI, we are able to outperform a baseline sequence-to-sequence model as well as Shen et al's [49] simpler model variant of averaging word embeddings to produce a sentence representation. Qualitative examples shown in in Table 8 and Supplementary Table 4 indicate that the model is able to capture some simple aspects of the mapping from sentiment or entailment labels to generated sentences/premises. There are however cases that may be attributed to a form of mode collapse where the model adds in trivial details to the caption to satisfy entailment, such as the color of one's shirt.

| Dataset | Sentiment Classification Accuracy | | | FPPL | | |
|---|---|---|---|---|---|---|
| | Positive | Negative | Overall | Positive | Negative | Overall |
| Senticap | 67.65% | 84.65% | 76.15% | 33.1 | 45.3 | 38.8 |
| Amazon Review | 78.29% | 54.68% | 66.48% | 14.2 | 14.7 | 14.4 |

Table 10: Sentiment Classification Accuracies using a trained FastText classifier [23] and FPPLs on 3200 generated samples from the SentiCap and Amazon Review datasets

# 6   Conclusion & Future Work

We investigate and demonstrate the potential of leveraging general purpose sentence encoders that produce fixed-length sentence representations to train generative models of sentences. We show that it is possible to train conditional generative models of text that operate by manipulating and transforming sentences entirely in the latent space. We also show that smooth transitions arise in the observed space when moving linearly along the input space of the GAN generator. In futre work, we we would like to evaluate our conditional text generation approach on a more challenging benchmark such as MultiNLI. Further, in our preliminary exploration of conditional text generation with SNLI, we experimented with a combination of MSE and adversarial training objectives similar to [39], but this showed no improvements over just adversarial training. However, ablating the coefficient of the adversarial term in this context, can shed some light on the impact of the autoregressive component and we hope to look into this in future work.

## Acknowledgements

The authors thank Alex Lamb, Rithesh Kumar, Jonathan Pilault, Isabela Albuquerque, Anirudh Goyal, Kyunghyun Cho, Kelly Zhang and Varsha Embar for feedback and valuable discussions during the course of this work. We are also grateful to the PyTorch development team [38]. We thank NVIDIA for donating a DGX-1 computer used in this work and Fonds de recherche du Québec - Nature et technologies for funding.

## Footnotes

[2]Official code from `https://github.com/jakezhaojb/ARAE`

[3]`https://github.com/salesforce/awd-lstm-lm`

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
