[Supplementary Material]

# Supplementary Material: Towards Text Generation with Adversarially Learned Neural Outlines

**Sandeep Subramanian**[1,2,4],[*] **Sai Rajeswar**[1,2,5], **Alessandro Sordoni**[4],
**Adam Trischler**[4], **Aaron Courville**[1,2,6], **Christopher Pal**[1,3,5]
[1]Montral Institute for Learning Algorithms, [2]Universit de Montral,
[3]cole Polytechnique de Montral, [4]Microsoft Research Montral,
[5]Element AI, Montral, [6]CIFAR Fellow

## 1  Appendix

We use this space to communicate additional results that could not be included in the main paper due to space restrictions. The results here help put results in the main paper in better context.

### 1.1  Unconditional Text Generation

In Table 1, we report statistics about the distribution of n-grams in the generated samples from our model as well as the WD-LSTM. We present the "unique n-gram ration" metric as a proxy for sample diversity which measures the ratio of unique n-grams to the real data measured with the same number of sentences (512,000). The "n-gram validity" metric which measures the percentage of n-grams in the samples that occurs in a holdout set of 6M sentences, is intended to serve as a proxy for sample quality. We see that when generating at low temperatures or with beam search, both models have extremely low sample diversity with our model having better n-gram validity. When sampling at high temperatures (i.e.) from the true distribution learned by these models, we can see that there is a significant difference between the diversity of our samples versus the WD-LSTM - the WD-LSTM actually contains more unique n-grams than the real data. Naturally, the n-gram validity metric captures this trade-off between sample quality and diversity of the two models.

A common criticism of GANs are that they appear to produce samples that are near copies of ones in the training set. We attempt to shed some light on the extent to which this happens in our model as well as the WD-LSTM. Specifically, we compute the tokenized BLEU-4 between samples and their nearest neighbor in the training set. We compute nearest neighbors using the FAISS[2] library in the space of averaged 300 dimensional GloVe vectors. The results appear to be dependent on the dataset, while our model overfits a lot more on the BookCorpus, it doesn't do so on WMT15 and less so than the WD-LSTM on SNLI.

### 1.2  Conditional Text Generation

In this section, we present some qualitative results we were unable to include the main paper. Specifically, we present samples from our model trained to generate sentences conditioned on sentiment labels in Table 3.

### 1.3  Walking the Space of Sentence Representations via Gradient Based Optimization

We include a few additional samples of walking the latent space of sentence vectors via gradient based optimization in Table 4.

---

[*]Work done while author was an intern at Microsoft Research Montreal
[2]https://github.com/facebookresearch/faiss

| Approach | Unique n-gram ratio | | | N-gram validity | | |
|---|---|---|---|---|---|---|
| | n=2 | n=3 | n=4 | n=2 | n=3 | n=4 |
| Ours (Beam) | 0.04 | 0.05 | 0.07 | 95.76% | 86.44% | 65.35% |
| WD-LSTM (Temp 0.5) | 0.05 | 0.07 | 0.12 | 95.22% | 84.87% | 61.68% |
| Ours (Temp 1.0) | 0.55 | 0.56 | 0.57 | 63.06% | 44.34% | 26.14% |
| WD-LSTM (Temp 1.0) | 1.38 | 1.30 | 1.29 | 46.20% | 31.75% | 17.52% |

Table 1: N-gram statistics of the generated BookCorpus samples from our model and a WD-LSTM. The Unique n-gram ratio metric is the ratio of the number of unique n-grams in our generated samples to the real data with an equal number of sentences. The n-gram validity quantifies the percentage of n-grams in the generated samples that are present in a holdout set of 6M sentences.

| Dataset | BLEU-4 | | | |
|---|---|---|---|---|
| | WD-LSTM | | Ours | |
| | 0.5 | 1.0 | Beam | 1.0 |
| BookCorpus | 26.8 | 6.8 | 47.7 | 17.92 |
| WMT15 | 16.6 | 4.6 | 13.7 | 5.5 |
| SNLI | 45.8 | 14.7 | 34.8 | 22.1 |

Table 2: A preliminary analysis of overfitting by presenting the average BLEU score between samples from the WD-LSTM and Our models with the nearest neighbor in the training set.

| **Positive** |
|---|
| i was very excited to read this book . |
| this is one of the best on the market . |
| if you are a fan of the music . |
| this book is one of the best things that i have ever read . |
| this is a very good product , and it is a very good way to not not do work . |
| a beautiful view of the beach . |
| a small crowd of people are walking on a beautiful beachfront |
| the girl is reading a newspaper in a nice , and white tiled room . |
| two little boys are having a good time with barbies |
| a happy woman standing in a large room filled with her computers |
| **Negative** |
| i bought this for my husband and had to replace it . the product is horrible . |
| i was so excited about this camera , but i was a little disappointed . |
| i would not recommend this book . |
| the product is a good product , but it did not work . |
| when i first got this item , i was very disappointed in the store . |
| a tired man with a machete |
| a wrinkled cat watching a dog . |
| a sad man is laying on the grass in front of a huge tree |
| a little girl is holding a dead bird in the water . |
| a dirty baby is jumping in the water |

Table 3: Samples from our GAN trained to conditionally generate sentences of a specified binary sentiment label on the SentiCap (bottom) and Amazon Review (top) datasets.

| Start | her boyfriend eyed him curiously and gave him a cautious nod , then followed her into the bar , not waiting for a reply . |
|---|---|
| 1 | her boyfriend eyed him curiously , not waiting . |
| 2 | her boyfriend gave him a cautious nod . |
| 3 | her boyfriend never showed up at the bar . |
| 4 | grace gave him a cautious nod of approval . |
| 5 | grace never met her a boyfriend that night . |
| 6 | her boyfriend never showed up at the mall . |
| 7 | grace never showed up at work that night . |
| 8 | grace never showed up at her job in the mall . |
| 9 | grace never showed up at her job in the mall that night . |
| 10 | grace never showed up at her job in the mall that started at 5:00 that night . |
| End | grace never showed up at her job in the mall that started at 5:00 that night . |
| Start | naturally , he was upset when i said it was over . |
| 1 | naturally , he was upset when i said it was over . |
| 2 | he was upset when she said it . |
| 3 | naturally , he was upset when she said it . |
| 4 | naturally , she was upset when she said it . |
| 5 | no way , she said . |
| 6 | no way , she said , turning slightly . |
| 7 | no way , she said , turning her head slightly so she was nose to nose with him . |
| End | no way , she said , turning her head slightly so she was nose to nose with him . |
| Start | " i 've been growing it out for a while . |
| 1 | " i 've been growing out of it . |
| 2 | " i 've been growing for a while . |
| 3 | " i 've been listening for a while . |
| 4 | " i 've been listening to a voice . |
| 5 | peter listen to me , you need it . |
| 6 | peter listen , you need to speak out . |
| 7 | peter listen , the voice didnt give him a chance to speak , you need to understand something . |
| End | peter listen , the voice didnt give him a chance to speak , you need to understand something . |
| Start | it 's one of the lesser-known failings of the vampire . |
| 1 | it 's one of the lesser-known failings of the vampire . |
| 2 | it 's one of the failings of the vampire . |
| 3 | im sure it 's one of the failings . |
| 4 | im sure it s one of their own . |
| 5 | im sure they got one of their own . |
| 6 | im sure theres a program of their own . |
| 7 | im sure they got a program of their own . |
| End | im sure cascadias got a program of their own , but it wouldnt be the same . |

Table 4: Examples of interpolations produced by our gradient-based technique on the BookCorpus.

|  | Latent Category 1 | Latent Category 2 | Latent Category 3 |
|---|---|---|---|
| 1 | she turned her head to look at him . | he did n't know how to stop him . | it was the most beautiful thing to do . |
| 2 | she had no idea what to do next . | he turned away from the window and left . | it was not a good thing to do . |
| 3 | but she knew it would be a lie . | he looked at her , his expression grim . | it had been a long time since then . |
| 4 | she did n't want him to hurt her . | he stopped and stared at the closed door . | it was n't fair to say that . |
| 5 | she asked , her eyes pleading with concern . | he looked at zane , who was smiling . | it was like a punch to the gut . |
| 6 | she wrapped her arms around him and squeezed . | he could n't help but smile at her . | it was a good idea to walk home . |
| 7 | her hand trembled slightly , her heart racing . | he wanted to know where he was going . | it was the strangest thing in the world . |
| 8 | she went to the bathroom to get dressed . | his voice was low , his eyes narrowing . | it was almost three o'clock in the morning . |
| 9 | she twisted her arms around his neck again . | he waited for a moment to regain consciousness . | it was a long way from the city . |
|  | **Latent Category 4** | **Latent Category 5** | **Latent Category 6** |
| 1 | i thought i was in love with him . | no , not your father , he said . | oh , shit , i 'm so sorry ! |
| 2 | i was in a hurry to find out . | i mean , you know what i mean . | come on , let 's go to bed . |
| 3 | i took a deep breath and exhaled slowly . | oh , yeah , well , i guess . | he asked , as if to say something . |
| 4 | i was pretty sure i was a fan . | look at me , i love you too . | she asked , her voice dripping with sarcasm . |
| 5 | i did n't mean to be with you . | i dont know , she said to herself . | she asked , her eyes wide with annoyance . |
| 6 | i heard the door slam shut behind me . | im sorry , she said with a sigh . | she asked , her eyes pleading with concern . |
| 7 | my heart was pounding , my chest hurt . | yeah , well , that 's all right . | oh , shit , she said to herself . |
| 8 | i got up and went to the kitchen . | when she did , she didnt say anything . | he said , his voice thick with emotion . |
| 9 | i swallowed hard and tried to sound stupid . | i mean , what are you doing here ? | yes , of course , i 'm sure . |
|  | **Latent Category 7** | **Latent Category 8** | **Latent Category 9** |
| 1 | " that 's what you 're doing here . | " that 's right , " he said . | " i love you , my love . " |
| 2 | " it 's all you have to say . | " i 'm fine , " she said . | " i 'm not a good man . " |
| 3 | " i do n't know where to go . | " i do n't , " ashe said . | " i do n't think so . " |
| 4 | " i did n't mean to upset you . | " i 'll go , " he says . | " i 'm sure it 's a trick . " |
| 5 | " you 've got to be kidding me . | " you 're beautiful , " she said . | " she did n't tell him that ! " |
| 6 | " she 's not going to kill you . | " it 's not , " he said . | " that 's why i 'm here . " |
| 7 | " that 's not what i 'm saying . | " you 're not , " he said . | " i 'll take care of you . " |
| 8 | " i 'm so glad you 're here . | " she 's not , " ethan said . | " maybe you can do that again ? " |
| 9 | " you 're going to be a hero . | " it 's okay , " i say . | " i thought you might like that . " |

Table 5: InfoGAN samples when varying a single categorical latent variable with 10 categories on the BookCorpus. The latent code captures significant factors of the text sentence such as the gender (1-4) of the subject(She/He/It/I), the presence of discourse markers (5), as well as other structures such as quotations (7,8,9). Different rows in each category correspond to different random samples of noise with the categorical latent variable fixed