[Reviews · NeurIPS 2018]

Reviewer 1



Update after author response: It's good to see the efforts of conducting human evaluation, and I encourage the authors to include more details, e.g., how the annotators are selected, what questions are asked, into the next revision. I'm changing the score to 5 now. ============== This manuscript proposes a generative adversarial approach for text generation. Specifically, sentences are encoded into vectors in a "latent space" by a RNN encoder; a decoder conditions on the vector and generates text auto-regressively. The latent representations are adversarially regularized towards a fixed prior. The method is evaluated on both unconditional and conditional generation settings, and the paper argues for better performance than baselines. The method is not that surprising, given Hu et al., (2017) (missing in the references), and Kim et al., (2017), among others. Many claims are made, including both theoretical justifications and better empirical performance. However, I do not find either of them is well-supported. I can hardly find any insightful theoretical discussion of the proposed method, in comparison to previous works. It is probably still okay, should the submission put more thoughts into the empirical part by, e.g., showing better performance in carefully designed human evaluation. Unfortunately, this is not the case. That being said, the third point of the claimed contribution --- the gradient-based interpolation between sentence embeddings appears intriguing to me. I would encourage the authors to discuss how it relates to existing works in more detail, and put it earlier in the narrative. In sum, this submission would have been much stronger should it come out, e.g., one year earlier. I cannot recommend that it is accepted to NIPS in its current status, but would be looking forward to a revised version. References Zhiting Hu, Zichao Yang, Xiaodan Liang, Ruslan Salakhutdinov, and Eric P. Xing. 2017. Toward Controlled Generation of Text. In Proc. of ICML.

Reviewer 2



The paper attempts to replace the MLE estimate for a sequence2sequence model with an adversarial criterion. The argument is that auto-regressive neural networks, i.e., recurrent neural networks, do not play nicely with adversarial training. The amount of novelty in the paper is limited as this paper is far from the first to consider a mix of recurrent neural networks and generation. Indeed, about a year ago Yoav Goldberg posted an adversarial review of a related paper: https://medium.com/@yoav.goldberg/an-adversarial-review-of-adversarial-generation-of-natural-language-409ac3378bd7. I think the evaluation of the paper is quite borked. How can you even consider an evaluation of non-conditioned generation? The paper is hopefully linguistically lost. As far as I can figure, the only reasonable way to evaluate unconditioned generation is through grammaticality, i.e., syntactic correctness. For instance, is the sentence “A yellow banana is lying on the table” better than the sentence “a purple banana is lying on the table” just because the former is more common, and, thus, rightly has higher probability under *any* good model? Of course not! Any attempt at computational creativity has to be able to distinguish likely (probably not creative) from syntactically valid. The paper ignores an entire field of linguistic inquiry. See, for instance, Lau et al. (2018) in Cognitive Science, who investigate the ability of neural language models ot assess grammaticality.

Reviewer 3



This paper proposed a new method for text generation. The groundtruth sentences are first feed to a pretrained encoder to obtain a fix length represetation. A GAN model is then trained to generate such represetations from a simple distribution (e.g., Gaussian). Finally, a decoder is trained to map the generated represeations to the words. The idea looks interesting and novel and experimental results are convincing to me . It would be better if the authors can further evaluate the effectiveness of using a pretrained sentence encoder. For example, what is the performance of using a variational autoencoder (maybe with pretrained encoder as initialization) instead of pretrained encoder with random noise? Minor: the proposed model is very similar to Sec 5.2 in https://arxiv.org/pdf/1806.02978.pdf, although that paper focused on a very different task. The only difference is the detailed model architeture. While I believe this model is developped independently, it would be good to add some discussion.